# A Queuing Network Model of a Multi-Airport System Based on Point-Wise Stationary Approximation

Xifan Zhao [1], Yanjun Wang [1,*], Lishuai Li [2,3] and Daniel Delahaye [4]

1 College of Civil Aviation, Nanjing University of Aeronautics and Astronautics, Nanjing 210016, China; zxf110300@nuaa.edu.cn
2 School of Data Science, City University of Hong Kong, Kowloon, Hong Kong SAR, China; lishuai.li@cityu.edu.hk
3 Faculty of Aerospace Engineering, Delft University of Technology, 2629 HZ Delft, The Netherlands
4 ENAC Lab, Ecole Nationale de L'Aviation Civile, 31400 Toulouse, France; daniel@recherche.enac.fr
* Correspondence: ywang@nuaa.edu.cn

**Abstract:** A multiple-airport system (MAS) consists of more than two airports in a metropolitan area under a large block of terminal airspace that is managed by one or two air traffic control units. When the capacity of an airport or of the terminal airspace drops, flight delays occur in the MAS system. A quick estimation and predication of traffic congestion in the MAS is important yet challenging. This paper aims to develop a queuing network model of MAS using point-wise stationary queues. The model analyzes the changes of non-stationary queues under the principle of flow conservation to capture flight delay propagation in the system. Regression analyses are performed to examine the relationship between the arrival and departure efficiencies of different airports. The model is validated with the data of Guangdong–Hong Kong–Macao Greater Bay Area airports. Simulation results show that the model can effectively estimate flight delays in the MAS.

**Keywords:** queuing network model; multiple airport system; delay propagation; M/G/1 system; airspace congestion

## 1. Introduction

Air transport is one of the fastest and most efficient modes of transportation among various methods of transportation, such as road transportation, sea transportation, and railway transportation. It is the primary choice for people in long-distance travel. As the International Civil Aviation Organization (ICAO) states, the air transport industry is more than a vital engine of global socioeconomic growth—it acts as a catalyst for economic development. With the continuous development of the air transport industry, the annual number of takeoffs and landings at airports and the number of passengers transported are increasing gradually. The increase in air traffic demand has driven economic growth and has simultaneously put enormous pressure on the air traffic management system. Frequent flight delays not only cause inconvenience to passengers but also introduce environmental problems such as increased fuel consumption and additional carbon emissions. According to the Civil Aviation Industry Development Statistical Bulletin of China, the national average passenger flight punctuality is 75.71%, while the average delay is 18 min per flight during 2015–2019. One of the most frequent discussions in air transportation research is to minimize flight delays through advanced technology and effective management [1].

Given the nature of flight operation, a flight delay that occurs at one airport propagates to other airports causing other flights to be delayed. Many studies have explored the problem of delay propagation in airport networks [2–4]. Factors that impact delay propagation have been investigated, such as the connection of flights in multiple airports and the use of common waypoints. Various methods have been applied to study flight delays from different perspectives. For example, machine learning techniques have been widely

applied to predict flight delays; complex networks approach has been used to uncover the fundamental properties of the delay networks; and operations research methods have been employed to minimize flight delays [1]. Bayesian networks have been widely used to investigate flight delays propagation in the airport networks [5]. Delay propagation in a network is mainly due to flight connectivity such as crew connectivity and passenger connectivity [3]. Research efforts have also been devoted to the development of arrival delay and congestion prediction models based on the Bayesian network [6].

Machine learning algorithms generally require a large amount of data and high computation accuracy. The reliability of prediction results is closely related to the categories and importance settings of influencing factors. Due to various uncertain factors affecting the operation of an air traffic system, timely estimating flight delay status under different operating scenarios is important. For example, given flight schedules and weather forecasts for the next 72 h, quickly predicting flight delays is important for the rational and effective implementation of traffic management strategies. Although fast simulations can simulate how the system will operate in the future, the results are highly dependent on the input parameters, the accuracy and precision of the simulation model, and the capability of the simulation system. The computational costs can be very high when a high-fidelity simulation is required.

Another widely used approach to estimating the state of a system is the development of a model based on queuing theory. Queuing theory has been applied to model many congestion problems. Early research mainly focused on the dynamic service process at single airports. Kivestu investigated the M/G/K queuing theoretical model for systems served by multiple servers simultaneously, where customers' arrival rates obey Poisson distribution, while service rates obey generally independent distribution [7]. One study demonstrated the derivation and application of the time constant of the M/G/l system, and proposed the concept of the delay algorithm. It has been proved that the algorithm can effectively simulate the dynamic process of the M(t)/Ek(t)/1 system. To simulate the approximate queuing process of flights in the airport, a delay model based on stochastic dynamic queuing theory and a Monte Carlo model was developed and validated with real data from two airports in New York: John F. Kennedy Airport (ICAO code: KJFK) and Newark Airport (ICAO code: KEWR) [8]. The point-wise stationary fluid flow approximation model was developed to determine the average queue length of a general arrival service distribution [9]. The model combines steady-state queuing and fluid flow modes to establish a linear differential equation for a single queue. Parameters of the service system under different service laws were analyzed. A method to calculate the queue length of different systems was proposed. Experimental results show that the model has good applicability to general queuing systems. An integrated surface–airspace departure model was developed to predict the time for an aircraft to taxi from the gate to the final departure point [10]. The model was further simplified into a relatively simple ordinary differential equation (ODE) based on the point-wise stationary fluid flow approximation. The movement process of the flight on the airport surface is modeled as a queuing network. Overall, the queuing theory has been demonstrated to be a very useful and suitable tool to simulate airport operations.

It is well known that flight delays propagate through airline networks and airport networks due to sharing of resources such as aircraft, crews, or airports. The approximate network delay model was originally proposed by Malone [11], and was further explored by Pyrgiotis et al., who developed an airport network delay (AND) model [2]. The key idea of the AND model is to capture the effects of primary delay on the overall network. The model consists of two modules: queuing engine (QE), which calculates the delay generated at each airport; the delay propagation algorithm (DPA), which updates the flight schedule and demand of all airports in the network. Although the AND model can effectively simulate flight delay trends, it does not consider en route congestion. In a recent study, a multilayer air traffic network delay model (multilayer air traffic network delay—MATND) was presented that captures the effect of the congestion in the air traffic network on flight

delays [12]. The MATND model consists of two parts: the air traffic network model and the stochastic dynamic queuing network model. The former is used to construct the air traffic network from the automatic dependent surveillance broadcast (ADS-B) data, and then the stochastic dynamic queuing model is used to calculate the delays caused by flights at each node and continue to track the impact on the subsequent servers. The stochastic model is based on the M/E/1 system, taking the airport and air congestion points as a single server, and uses non-stationary arrival and k-Erlang distribution to simulate flight operations. The model can estimate queuing situation at each node at different times. The MATND model is validated and shows good performance in quantifying delays in China's air traffic network.

In addition to the studies of flight delays in an individual airport or in airport networks, there has been extensive work on multi-airport system (MAS) operations. Among the many MASs worldwide, the New York MAS, the San Francisco Bay Area MAS, and the Los Angeles MAS have been widely investigated [13]. The research has focused on airspace operation [14], traffic flow pattern recognition [15–19], departure metering [20], and capacity improvement [21–23]. However, little research has been performed on the estimation of flight delays in a MAS. Given their geographical locations, the operations of each airport in a MAS are interdependent. Modeling and estimating flight delays in a MAS is challenging. In this paper, we present a flight delay estimation model for a MAS that considers the characteristics of airport and airspace operation based on queuing theory. The overall modeling framework is shown in Figure 1. The virtual airport in the figure represents all other airports except the ones in the MAS. Our model uses stable fluid flow to estimate the average length of queues with general service distribution characteristics and Poisson arrival distribution. It can track the propagation of individual server delays in the MAS. We validate our model with real data from the MAS in Guangdong–Macao–Hong Kong Great Bay Area (GBA). The main contribution of this paper is the development of a queuing model for a MAS considering terminal airspace constraints. We explore the correlation between the service efficiency of inbound and outbound flights from different airports in the MAS. The model can serve as a decision support tool for the traffic managers to alleviate airspace congestion and improve the operational quality of the MAS.

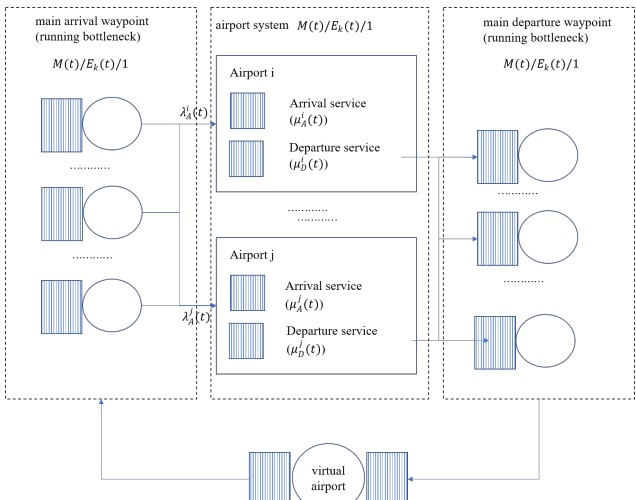

**Figure 1.** Schematic diagram of a queuing-based model of a multiple airport system. The model of the MAS system is divided into four parts: the arrival waypoints, the departure waypoints, the airports in the MAS, and other airports.

## 2. The Queuing Network Model for the MAS

Here, we discuss the development of our queuing network model for the MAS in GBA. The data used to construct the model are from two sources: empirical flight data and airspace operation data. The model can explore the impact of flight congestion generated

by individual servers (e.g., an airspace route point in the MAS) on the MAS. The local queue lengths of each server including the waiting time are computed by taking upstream servers into account. The MAS network model proposed in this paper is based on the single-queue equation of point-wise stationary approximate fluid flow. The airports and airspace congestion points are assumed to be single-server queuing systems. By combining with approximate queuing and fluid flow methods in stationary state, the model can capture the dynamic process of flights entering and leaving the queue through sliding the time window. Figure 2 shows the flow chart of the model. The dashed line box in the figure is the input of the model, where the parameters are obtained from the input flight data. The changes of the queue at each server are computed in every five-minute intervals starting from $t_0$.

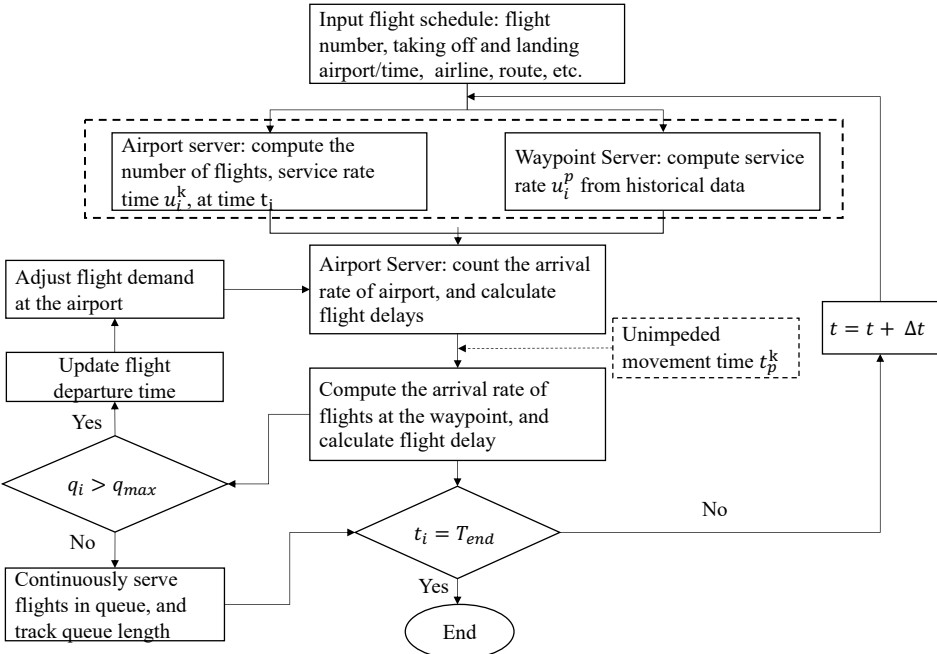

**Figure 2.** The overall flow chart of the model.

### 2.1. Single-Queue Model

For a single-server queuing equation with non-stationary dynamics, let us assume that the average queuing length at time $t$ is $x(t)$. Then, the flow change at $t$ is $x(t) = dx(t)/dt$. According to the principle of flow conservation, the change of the queue is equal to the difference between the inflow and outflow of the queue at time $t$. Denote the outflow and inflow in the queue at time $t$ by $f_i(t)$ and $f_o(t)$, respectively, then we have:

$$\dot{x}(t) = -f_o(t) + f_i(t) \tag{1}$$

The outflow can be expressed as $f_o(t) = \mu(t)\rho(t)$, where $\mu(t)$ denotes the service of queue at time $t$, $\rho(t)$ is the utilization rate of the service rate. In the case of no limit on queue length, the inflow can be expressed as $f_i(t) = \lambda(t)$, where $\lambda(t)$ is the arrival rate at time $t$. Equation (1) can be rewritten as:

$$\dot{x}(t) = -\mu(t)\rho(t) + \lambda(t) \tag{2}$$

The expression of $\rho(t)$ in Equation (2) depends on the service rate distribution of the queuing systems. Determining the exact expression for $\rho(t)$ is difficult even for the simplest queues. Therefore, we employ a point-wise stationary method to determine the utilization rate. Here, the average queue length is given as $x = G_1(\rho)$. Then, the average utilization rate can be obtained as $G_1^{-1}(x(t))$. The queuing system used in our model is

M/G/1, where the arrival process is subject to Poisson distribution, and the service time is subject to arbitrary distribution. According to the Pollaczek–Khintchine (P–K) formula, the average queuing flight of the steady-state system is:

$$x = \rho + \frac{\rho^2(1 + C_v^2)}{2(1 - \rho)} \tag{3}$$

where $C_v^2$ is the square of the coefficient of variation of service time. The corresponding expression of $\rho$ with respect to x is:

$$\rho = \frac{x + 1 - \sqrt{x^2 - 2C_v^2 x + 1}}{1 - C_v^2} \tag{4}$$

By substituting Equation (4) into Equation (1), we obtain the approximate equation of the point-wise stationary fluid flow for the single-column M/G/1 queuing system:

$$\dot{x}(t) = -\mu(t)\frac{x(t) + 1 - \sqrt{x(t)^2 - 2C_v^2 x(t) + 1}}{1 - C_v^2} + \lambda(t) \tag{5}$$

### 2.2. The Queuing Network Model of a MAS

We extend the above single-queue model to a multi-queue network model based on the principle of flow conservation. We connect the servers (i.e., airports, or waypoints) according to the flight path. The input and output of each service system are obtained from empirical flight data. The output of the upstream server is used as the input of the downstream server. The average queue length of each server in the network model is then given as follows:

$$\dot{x}(t) = -\mu_i(t)\frac{x_i(t) + 1 - \sqrt{x_i(t)^2 - 2C_{vi}^2 x_i(t) + 1}}{1 - C_{vi}^2} + \lambda_i(t) + \sum_j f_{ji}(t - \tau_{ji}) \tag{6}$$

where $f_{ji}(t - \tau_{ji})$ represents the number of flights entering queue $i$ after leaving queue $j$ at time $t - \tau_{ji}$. $\tau_{ji}$ is the unimpeded movement time between servers $i$ and $j$. $\lambda_i(t)$ is the inflow that has not been served by other servers before entering queue $i$ (in this paper, it is the traffic flow outside the MAS or the number of flights departing from the airport). Therefore, the inflow of servers in the queuing network can be grouped into two categories: (i) the outflow that has been served by the upstream server; and (ii) the external input that has not passed through other queues and does not contain the queuing waiting time of non-local servers. The network model established by Equation (6) includes the waiting time of the flight in the upstream server and the time consumed by the service process when computing the waiting time of each flight. It can compute the time of each flight when it enters the queue of a server and the time when leaving the queue. Thus, we can calculate the delay time of each flight. The local delay that occurred at each server is guaranteed to be propagated to the downstream server along the flight's path.

To illustrate how the model works, we plot a simplified queuing network model in Figure 3. The model has five independent servers, with three airports, and two waypoints. Assume that a flight departs from its origin airport to route points servers $A$ and $B$ at time $t$. We can see that number of the flights $i$ and $j$ flow to service system $A$ and $B$ through original airport. The unimpeded flight times from departure airport to service systems $A$ and $B$ are $\tau_i$ and $\tau_j$, respectively. In addition, the traffic $\lambda$ from other airports (flow from other airports or route points other than the five servers in the figure) arrives at service system $A$ at time $t$ and enters the queue to wait for service. The calculation of the dynamic queue length changes for servers $A$ and $B$ are as follows:

$$\dot{x}_A(t) = -\mu_A(t)\frac{x_A(t)+1-\sqrt{x_A(t)^2-2C_{vA}^2 x_A(t)+1}}{1-C_{vA}^2} + \lambda(t) + f_{oA}(t-\tau_i) \quad (7)$$

$$\dot{x}_B(t) = -\mu_B(t)\frac{x_B(t)+1-\sqrt{x_B(t)^2-2C_{vB}^2 x_B(t)+1}}{1-C_{vB}^2} + f_{oB}(t-\tau_j) \quad (8)$$

The service rate for the servers $A$ and $B$ are $\mu_A$ and $\mu_B$. $C_{vA}$ and $C_{vB}$ are the coefficient of variation of the service rate, respectively. The difference between the calculation of the two queue lengths is linked to the flow entering the queue as external input $\lambda$ into server $A$. In the following section, we discuss how to obtain the service rate, and traffic arrival rate from empirical flight data.

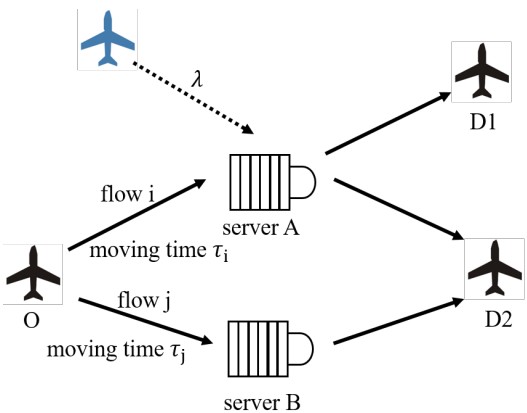

**Figure 3.** An example showing traffic flow between servers.

*2.3. Data Processing and Analysis*

2.3.1. Airport Service Rate

The main input of the model is demand rate, service rate, and the route of flights. Parameters such as demand rate and flight route can be obtained directly through flight schedules and historical flight information. Service rates of airports and main route points need to be processed through historical data. First, the entire day from 0:00 to 24:00 will be divided into 280 equally time windows, $t_i$. Let $i$ be the $i$th five minutes window of the day. The service rates $\mu$ and arrival rates $\lambda$ are assumed to be constant within each time window.

To compute the service rate of an airport, we should consider two factors: the number of incoming flights and the impact of weather. Flight data operated during extreme weather was cleaned out. We select the date when the number of departing flights reaches at least 70% of the maximum departures. Then, we compute the number of arrival and departure flights in every five minutes, such that the trade-off between departures and arrivals is captured.

Figure 4 presents an example of the computation of average service rate for a certain time window. Figure 4a shows the service rate of different percentiles when there is only one flight departing from the airport. The red line is the 85th percentile of the data. We found that the corresponding service rate when selecting the 85th percentile is close to the actual service efficiency at the airport. Thus, it is taken as the final service efficiency under this condition in the model. Figure 4b shows the service rate and its linear fitting results under different numbers of incoming flights. The time period selected is an off-peak period, when the operating efficiency of the airport has not yet reached saturation. It is not difficult to find from the figure that as the number of arrival flights increases, the service efficiency of the airport for departing flights gradually decreases. This indicates that the arrival flights have higher priorities than the departing flights.

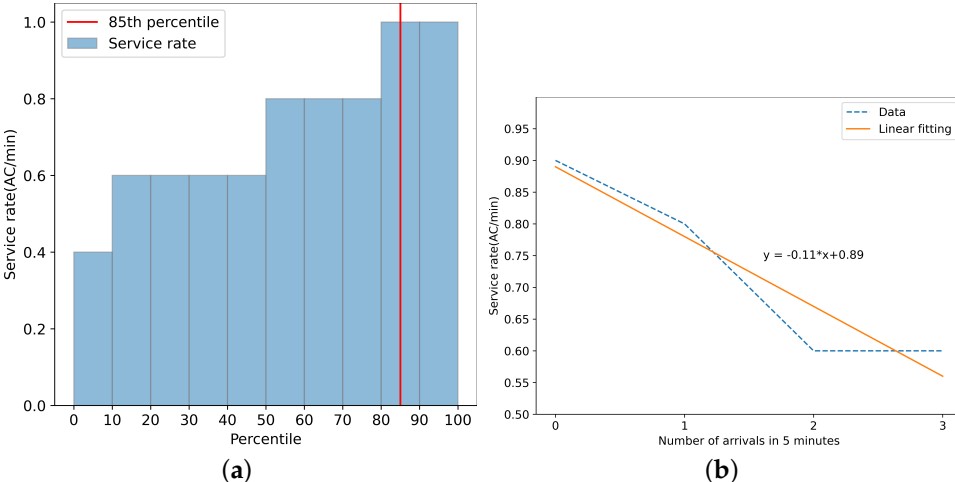

**Figure 4.** Service rate computation. (**a**) Average service rate (five minutes); (**b**) Service rate estimation.

#### 2.3.2. The Correlation between Airport Service Rates in the MAS

The airport capacity envelope, as a representation of airport capacity, is widely used to study the relationship between airport departure rates and airport arrival rates. It has been extended to analyze the relationship between arrivals and departures between different airports in a MAS. To capture the relations between departures and arrivals, quantile regression is used to estimate the linear relationship between the two variables. Let $\tau$ be the probability of a data point in the dataset that is less than or equal to a certain value $y$, i.e., $\tau$ quantile of $y$, for which the general expression is $\tau = P(Y \leq y(\tau))$. When using quantile regression to analyze the correlation of traffic flows at airports $i$ and $j$, it is necessary to set the number of quantile points to minimize the residuals. The number of arrival flights at airport $i$ and the number of departing flights at airport $j$ are calculated. The following segmented linear functions is established.

$$R_\tau\left(y^j \mid x^i\right) = \alpha_k x^i + \gamma_k, x^i \in \left[0, 1, 2 \ldots x_{\max}^i\right] \tag{9}$$

where $x^i$ and $y^i$ are the numbers of arrival (departure) flights at airport $i$ and $j$. The range of $x^i$ is from 0 to the maximum value of the number of arrivals (departures) per unit of time in the airport $i$; $\alpha_k$ and $\gamma_k$ represent the slope and intercept of the $k$th function, respectively. A linear programming model is established to minimize the residuals. Please refer to [24] for more details.

In the MAS we studied, Guangzhou Baiyun Airport and Shenzhen Baoan Airport are the two hub airports that are about 120 km apart. Figure 5 plots the envelope diagrams of arrival–departure envelops for the two airports. The larger the point, the more occurrences of the same number of arrivals and departures. It can be seen that when the arrival capacity of Shenzhen Airport is saturated, the departure efficiency of Guangzhou Airport is about 12 flights per 15 min. The maximum departure efficiency of Guangzhou Airport is  flights per 15 min. However, due to the influence of Shenzhen Airport, the number of departures at Guangzhou Airport is reduced to 12 flights per 15 min. It is about 66.7% of the efficiency of its full departure. It can be seen that there is a clear downward trend but the efficiency decreases slowly. With the increase in arrivals at Shenzhen Airport, the number of departures at Guangzhou Airport has decreased significantly but relatively slowly. All the envelope shapes in the figures are rectangles except for Figure 5a. The rectangle in the regression model indicates that there is no close relationship between the regressor and the explained variable. The possible reason is that although the distance between the two airports is close, the destinations of the flights do not overlap with each other, or the shared used airspace is not saturated. Generally speaking, when flights from different airports need to pass through a shared waypoint, they may have a restrictive effect on the flights

departing from other airports. They will have a higher impact on each other due to airspace capacity constraints at peak hours especially.

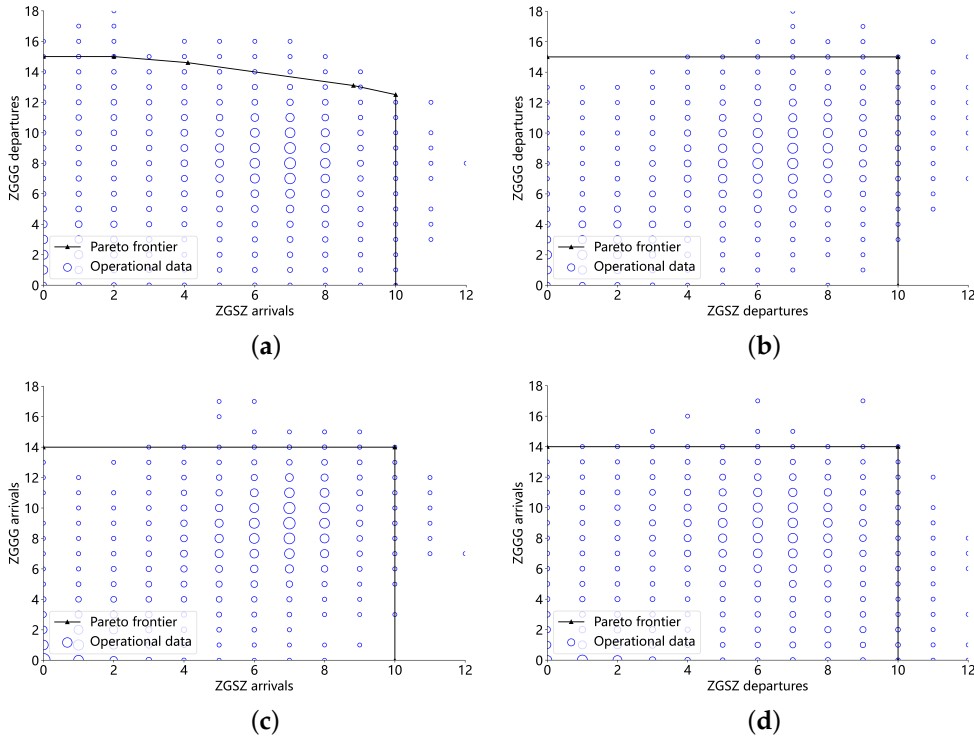

**Figure 5.** Airport capacity coverage curves. (**a**) Shenzhen (arr)–Guangzhou (dep); (**b**) Shenzhen (dep)–Guangzhou (dep); (**c**) Shenzhen (arr)–Guangzhou (arr); (**d**) Shenzhen (dep)–Guangzhou (arr).

### 2.3.3. Unimpeded Movement Time between Servers

When calculating the queue of flights entering and leaving different servers, it is necessary to calculate the unimpeded movement time between two nodes. In this study, the time refers to the flying time between the airport and the waypoint. The time for flights flying between an airport and a waypoint may vary in a certain range. Large deviations from the nominal flying times are generally caused by the maneuver clearance given by air traffic controllers to avoid weather or conflict. To account for such variations, the flying time can be modeled as two parts: the free movement time and queuing time. However, tested results show that the variation of flying time has little impact on the model computed flight delay. Thus, the flying times between the airport and waypoint is set to be a constant value in the model.

## 3. Results

We apply the queuing network model to study the MAS in Guangdong–Hong Kong–Macao Greater Bay Area. There are five airports in the MAS which are Guangzhou Baiyun International Airport (ICAO code: ZGGG), Shenzhen Bao'an International Airport (ICAO code: ZGSZ), Zhuhai Jinwan Airport (ICAO code: ZGSD), Huizhou Pingtan Airport (ICAO code: ZGHZ), and Macau Airport (ICAO code: VMMC). Hong Kong airport is not considered because its traffic flow has been separated from these five airports. The two main airspace route points YIN and LMN are identified by the air traffic control authorities that have a huge impact on the whole MAS operation. We obtained the 2019 flight data to calculate the parameters of the model. Figure 6 shows the geographical positions of the airports and route points. The gray line is the boundary of the terminal airspace of the MAS.

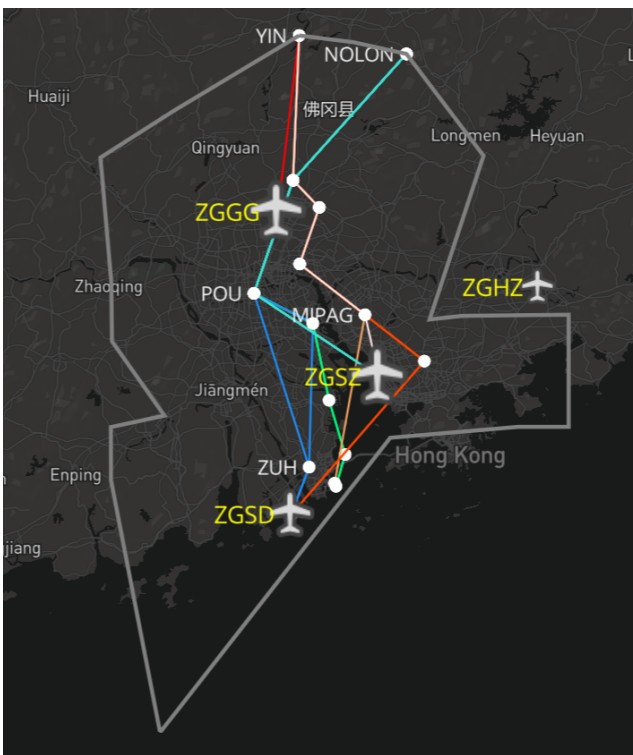

**Figure 6.** Location of airports in Guangdong–Hong Kong–Macao Greater Bay area.

### 3.1. Time Average Queue Length

Figure 7 depicts the departures demand and service rate every 15 min throughout the day at Guangzhou Airport and YIN route point. We can see that the demand and service rates for flights suddenly increase from 06:00 and gradually decrease after 20:00 for both the airport and the route points.

Figure 8 presents the estimation results by the model for each airport and waypoint. The orange line is the actual number of flights departing every 5 min in the queue, and the blue line is the model estimation result. It can be seen that the two lines fluctuate to a similar degree. There is a good match between the queue lengths derived from the model and the actual results. The model captures the actual departure situation and can reflect the possible delay peak period.

The difference between the model estimated results and the actual ones may be due to the fact that the sequence of flights entering the queues in the model is determined by the flight schedule on a first-come-first-served (FCFS) basis. Taxiing on the airport surface is not considered in the model. In practice, the air traffic controller can adjust the sequence of departure/arrival according to the situation at the airport. This may lead to the deviations of modeled queue length from the actual queue length. In addition, the model operation effect of the route point server is slightly less accurate. This is because of the uncertainties of flying times for flights from the airport to the route points. We chose the medium of flying times from historical data as free movement time between queuing systems. In practice, the scheduled flight time between airports is usually composed of the minimum flight time plus a certain amount of redundant time. The uncertainty of flight time can lead to a bias in the delay estimation of the route point queuing system. In addition, flights passing through a designated route point are usually affected by the service efficiency of the upstream server. The queue length, therefore, includes the waiting time at the upstream server, which is one of the reasons for the inconsistency between the estimated and actual results.

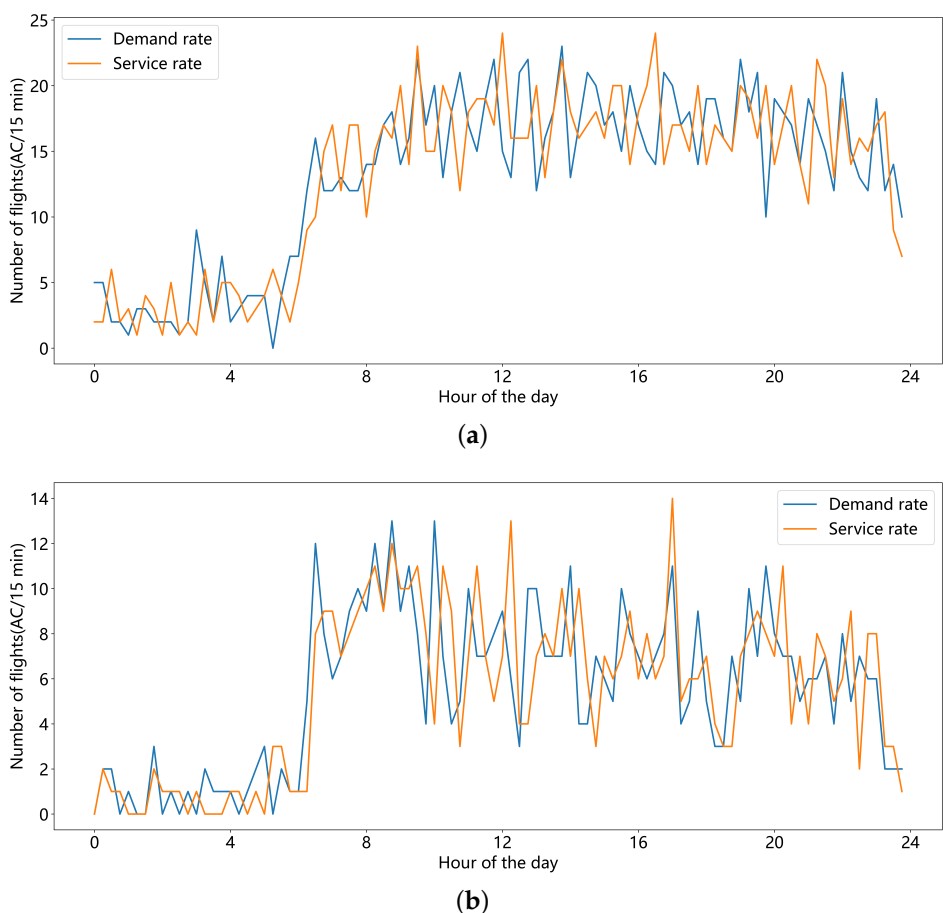

**Figure 7.** Demand and service profile at airport and waypoint. (**a**) Demand and service profile at ZGGG; (**b**) Demand and service profile at YIN.

The model proposed in this paper does not consider flight cancellations. The daily takeoffs and landings at Huizhou Airport are relatively small. There is a capacity constraint at this airport. Major flight delays were caused by upstream airports or route points. Thus, the operation of this airport is considered to has little impact on the whole MAS operation.

Two parameters are directly related to the reliability of the estimation results: the service efficiency and flight arrival distribution of the server. It can be seen from Figure 8 that at the beginning and end of the day the estimated results of the model are not much different from the actual data. The model can capture the changes in the number of flight departures accurately; this is because flight arrivals and service efficiency during off-peak periods are less affected by external factors. The amount of tactical adjustments during real operation is small. When congestion occurs in the airport or airspace during peak hours, tactical traffic controls depend on the traffic situation. There is a certain gap between the model parameters obtained through historical data and the actual operation. In addition, we found that the model curve is relatively stable with few upward and downward fluctuations. This is because there are no abrupt changes in service rates of the adjacent time windows in the model. Even if it is affected by the queue length, the service rate changes slowly. In general, the model can still capture the overall operation of the MAS and can estimate flight delays in the MAS.

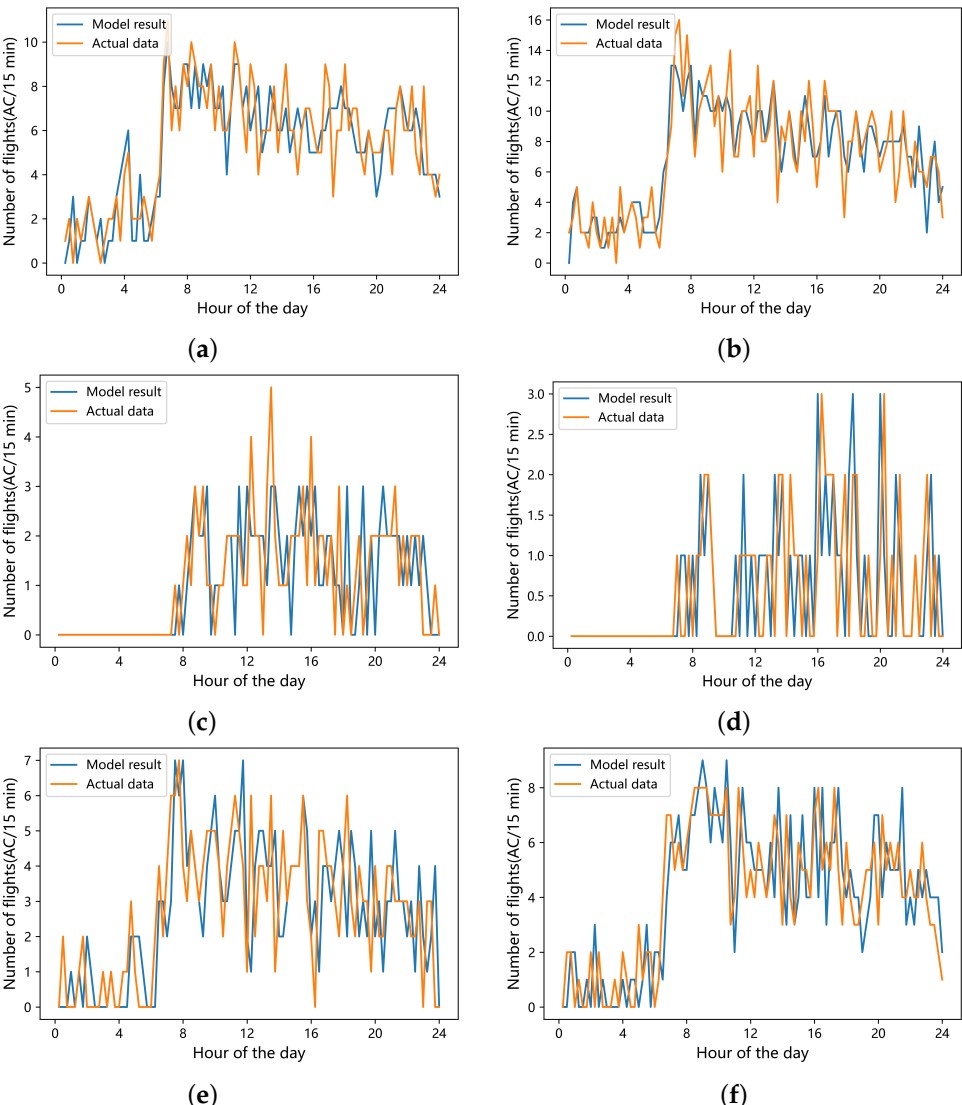

**Figure 8.** Hourly departure traffic at the four airports and two waypoints (LMN and YIN). (**a**) ZGSZ; (**b**) ZGGG; (**c**) VMMC; (**d**) ZGSD; (**e**) LMN; (**f**) YIN.

### 3.2. Average Delay Time

Figure 9 shows the average hourly delays at Guangzhou and Shenzhen airports in November and December 2019. The blue curve in the figure is the real average hourly delay, while the shadow covers the 95% confidence interval of the delay. The orange line is the estimated delays on 21 December in 2019 by the model. It can be seen from the figure that there is a slight discrepancy between the model estimated results and the actual delay. The fluctuation trends of the two lines are the same.

Figure 10 compares the actual average delay and the average delay estimated by the model for each airport and route point. Overall, the model can effectively estimate the average flight delay. The average delay time estimated by the model is slightly different from the actual one, partly due to the interference of other factors during real operation, such as tactical adjustments. Compared with other airports, the gap between actual and estimated delays of Macau Airport are bigger. This is because the number of flights in this airport is relatively small, and there are not many flights delayed due to the sufficient airport capacity; flight delays are mainly caused upstream delays.

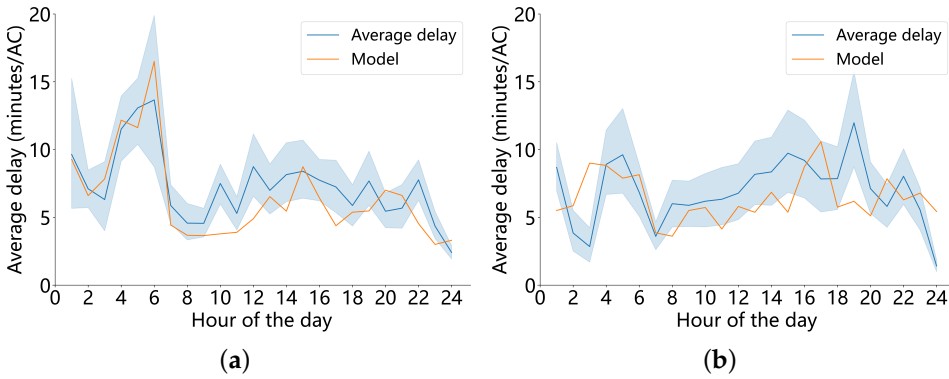

**Figure 9.** Hourly delay computed from 2-month historical data and simulated data. (**a**) Average hourly delay at ZGGG; (**b**) Average hourly delay at ZGSZ.

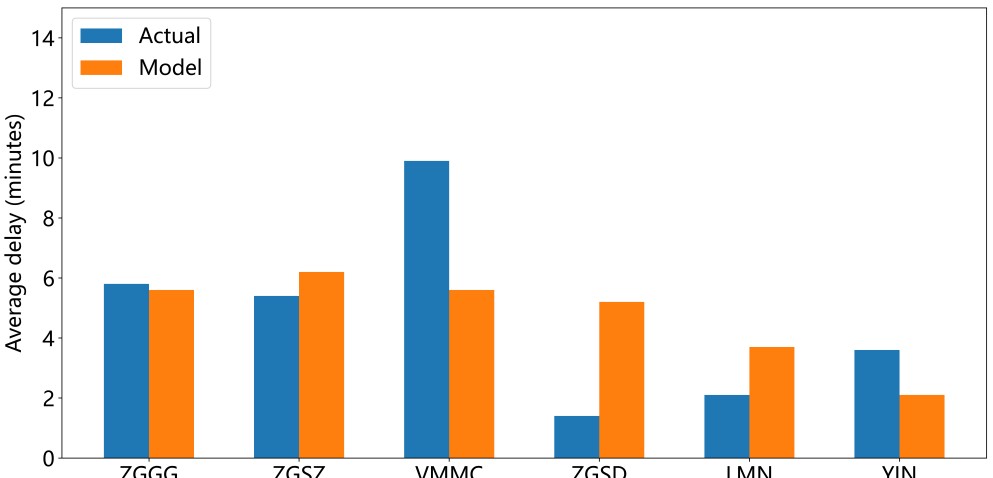

**Figure 10.** Average flight delay at the airports and waypoints.

We observed that there is notable difference between the actual delay and estimated delay at ZGSD as well. One possible reason is that the service rate for the model is lower than the actual service rate. The maximum number of hourly scheduled flights at ZGSD is 20, which is much smaller than the capacity of a single-runway airport. Given the limited information, we cannot adjust service rate in the model manually. Second, we found in the operational data that a certain number of flights were taking off earlier (5–10 min) than the scheduled time, while the flight delay is generally caused by the upstream delay. Therefore, the averaged actual flight delay is very low. The estimated results could be improved if more detailed information were provided.

The modeling results for the airports are better than the route points, which is consistent with the queue length estimated results. The main reason is influenced by the uncertainty of the flying time between airports and route points.

## 4. Conclusions

This paper proposes a queuing network model for a multi-airport system (MAS) which aims to estimate the queue length and flight delays at airports and route points. The model was validated through flight data in 2019 of the Guangdong–Hong Kong–Macao Greater Bay Area MAS. The results show that the point-wise stationary network model can effectively capture the phenomenon of flight congestion and delay propagation. The model can be used as a tool for flight schedulers and traffic managers to quickly estimate flight delays. Compared with the high-fidelity simulation tools, the main advantage of the model is that it is very easy to use and can quickly calculate flight delay and traffic values between airports and waypoints. The model can support traffic managers in various

ways. First, it can quickly compute flight delays and traffic throughput at the airport and waypoint once the traffic demand and service rate are input. The estimated traffic situation helps traffic managers to maintain situation awareness, such that appropriate traffic management initiatives and resource allocation plans can be prepared in advance. For example, preparations can be made if a thunderstorm is predicted at the airport next day. Airlines, airports, and air traffic control authorities work together to determine the plan to reduce flights to balance traffic demand and capacity. When a schedule is decided, all stakeholders can be aware of the estimated traffic situation. Second, the model can help traffic managers to test "*what-if*" scenarios. Since the operations of airports in the multiple airport system are correlated, the change of airport capacity at one airport or at a waypoint may have unanticipated impact on the performance of the whole system. The slot allocation department can use the model to test various slot allocation schemes when the capacity is changed.

There are some limitations to this study, as all nodes (airports and route points) are assumed to be served sequentially by a single server system on a first-come-first-served basis. When there are multiple servers (such as multiple runways), the presented model cannot simulate the operation of flights on the airport surface. It will result in differences in the arrival queue order and departure time of flights, thus causing the difference between the estimated and actual results. Therefore, future work could investigate queuing models that include airport surface operations and introduce real-time calculations to more accurately predict delay peak adjustment decisions. In addition, the direction of airport runway operations and other user activities in the airspace are also important factors affecting flight operations. All of these factors can be implemented in further refining the model.

**Author Contributions:** Conceptualization, Y.W.; data curation, X.Z.; formal analysis, X.Z., Y.W., L.L. and D.D.; software, X.Z.; writing—original draft preparation, X.Z. and Y.W.; writing—review and editing, Y.W., L.L. and D.D. All authors have read and agreed to the published version of the manuscript.

**Funding:** This research was supported by the National Natural Science Foundation of China (Grant Nos. U2033203, U1833126, 61773203, 61304190).

**Institutional Review Board Statement:** Not applicable.

**Informed Consent Statement:** Not applicable.

**Data Availability Statement:** The data used to support the findings of this study are available from the Central-south Air Traffic Management Bureau, Civil Aviation Administration of China, upon request.

**Conflicts of Interest:** The authors declare no conflict of interest.

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
