# Peer review of "A Queuing Network Model of a Multi-Airport System Based on Point-Wise Stationary Approximation"

_aerospace, doi:10.3390/aerospace9070390_

Round 1

Reviewer 1 Report

1.    In section 2.2, the proposed queuing network model of a MAS employed flight times τi andτj as constant values even though there exists large variations in their flight times. Generally speaking, the airport network models define the flight times as constant values like the authors selected, but is this assumption grasp the airport network well? Please discuss the point.

2.    In section 2.3.1, the authors calculated the arrival and departure rate in every five minutes. In section 2.3.3, the authors assumed the flight time between nodes were very small, but it might be non-negligible under the five minutes time intervals. Please clarify how the assumption effects on the network analysis.

3.    In section 3, the authors mentioned that the gap between actual and estimated delays of Macau Airport is bigger in Figure 10. In Figure 10, the gap corresponding to ZGSD is also big. Please discuss it.

4.    In section 1 line 119-120, the authors stated that “the model can serve as a decision support tool for the traffic managers”, but it was unclear how this model support their decisions. Please discuss how this model contributes as a decision support tool.

Reviewer 2 Report

The submitted paper is an interesting attempt to model airport services by queueing methods. The kay equations are (6)-(8) which would give us the measures. However, I could not see the solution to these systems.

Furthermore, the authors admitted that their approach is only for single-server systems, thus it is not realistic.

From queueing  theoretical point of view I do not see any novelty in the paper, the applied mathematical level is average.

I cannot judge the practical importance of the paper.

A major revision is needed for the acceptance, answering my questions concerning the above mention equations.

Reviewer 3 Report

This is a nice piece of research. I think that the topic of flight delays will in future become more and more important because of limited airport capacity at hubs (after air traffic has recovered from the recent crises).

The model is easy to implement, but is not as precise as more advanced methods, as the authors state. I think this is OK and the model has its own merits. The authors are very clear about this. However, I'm not an expert in queuing models, but it is well explained.

I have just a few minor remarks:

- line 66: these are IATA, not ICAO codes

- line 359: a typo (test - scenarios)

There is some yellow highlighted text, I don't know if that is intended.

Author Response

Thank you very much for your comments and suggestion. We have corrected all the typos and grammar mistakes in the revised manuscript. We provided point-to-point responses to your comments.  For review convenience, we have highlighted (in Yellow) any additions, edits, and changes in the revised manuscript.

P1. - line 66: these are IATA, not ICAO codes

Response:  Thank you for pointing it out. We have changed the IATA codes to the ICAO codes, i.e. JFK to KJFK, EWR to KEWR.

P2. - line 359: a typo (test - scenarios)

Response: Thank you. The text should be "test 'what-if' scenarios". 

Round 2

Reviewer 2 Report

The authors wrote their answers, but from a mathematical point of view, I cannot accept them.

Please ask for a new review from an expert in this field because the paper is not about mathematical queuing theory.

Author Response

We thank the reviewer for the comments. Our work focuses on applying queuing theory to the air transportation field. The development of queuing theory is out of the scope of this study. Thanks again for your time and suggestions.